# Reproductive Biology and Breeding Systems of Two *Opisthopappus* Endemic and Endangered Species on the Taihang Mountains

**DOI:** 10.3390/plants12101954

**Published:** 2023-05-11

**Authors:** Yiling Wang, Yafei Lan, Hang Ye, Xiaolong Feng, Qiyang Qie, Li Liu, Min Chai

**Affiliations:** 1School of Life Sciences, Shanxi Normal University, Taiyuan 030031, China; 220112015@sxnu.edu.cn (Y.L.); 221112089@sxnu.edu.cn (X.F.); 221112090@sxnu.edu.cn (Q.Q.); 20210095@sxnu.edu.cn (L.L.); 2Key Laboratory of Resource Biology and Biotechnology in Western China, Ministry of Education, College of Life Sciences, Northwest University, Xi’an 710069, China; phdye@stumail.nwu.edu.cn

**Keywords:** Floral syndrome, Protandry, Pollination, Mating systems

## Abstract

*Opisthopappus* is a perennial, endemic herb of the Taihang Mountains in China. Two species of this genus (*O. longilobus* and *O. taihangensis*) are important wild genetic resources for Asteraceae; however, their reproductive biology has been lacking until now. This study is the first detailed report on the reproductive biology and breeding systems of two *Opisthopappus* species. Through field observations, the floral syndromes of *O. longilobus* and *O. taihangensis* were found to possess a similar pattern, although *O. taihangensis* has a relatively larger capitulum, more ray ligules, and disc florets. The flowers of both *O. longilobus* and *O. taihangensis* are protandrous, a character that can prevent autogamy at the single-flower level, and insects are required for pollination. Further, brightly ligules, brightly bisexual florets, unique fragrance, and amount of nectar suggest that these species propagate via an entomophilous pollination system. Hymenopteran and Diptera species were observed as the effective pollinators for these two species. The outcrossing index, pollen/ovule ratio and the results of hand pollination indicated that these *Opisthopappus* species might have a mixed mating system that combines cross-fertilization and partial self-fertilization for *O. longilobus* and *O. taihangensis*, outcrossing predominated in the breeding system, while self-pollination played an important role in seed production when insect pollination was unavailable, particularly in a harsh environment, such as the Taihang Mountains cliffs. Meanwhile, *O. taihangensis* might better adapt to severe surroundings with relatively complex floral syndromes, specifically through the attraction of visiting insects and a high seed set rate. The above results not only provide reference information toward a better understanding of the survival strategies of *O. longilobus* and *O. taihangensis* in the Taihang Mountains but also lay a solid foundation for further exploring the molecular mechanisms that underly their adaptation under cliff environments.

## 1. Introduction

Reproduction is not only the most important and relatively fragile step in the life cycles of plants but also the core of their evolutionary process [1,2,3,4]. Meanwhile, pollination biology is a critical parameter in the life history of flowering plants that determines the success of sexual reproduction [5,6,7]. The mating and breeding systems of plant species are determinant factors of the genetic structures of populations [7,8]. Thus, an investigation of the characteristics of the reproductive biology of a species, including their pollination ecology and breeding system, is indispensable for elucidating the different phenophases of a species. Further, to explore the strategies they employ to adapt to changing habitats, particularly for species that thrive in harsh environments [4,6].

Cliff habitats are a typical type of harsh environment. where cliff-dwelling plants not only experience an extreme and harsh climate but are also subjected to harsh environmental pressures and ecological constraints [9,10]. Plants that grow in such habitats are less likely to disperse pollen and seed over long distances. Cliff habitats are sensitive to climate change, which can disrupt the overlap in seasonal timing of flower production and pollinator activity, which further reduces pollination [11,12,13]. Thus, the reproductive success of these lithophytes may be reduced by both uncertain pollinator services and low pollen outputs [4,14]. Studies of the reproductive characteristics of these lithophytes can improve our understanding of their evolutionary processes and survival strategies, yet there have been few corresponding investigations to date [3,6,7,15]. 

The Taihang Mountains (36–40° N, 112–115° E) lie between the Ordos-Shanxi Plateau and North China Basin, which is a prominent natural boundary in Northern China [16,17]. With an intense uplift during the Late Pliocene to Pleistocene, this mountain range forms a very complex topography, having many gullies, valleys, cliffs, and slopes, which provide unique and diverse environments in different regions [18,19]. Thus, several rare and endemic plants grow in this area, including the *Opisthopappus* species [19,20]. However, little is known regarding the reproductive characteristics and breeding patterns of these endemic plant species [17,21,22].

The *Opisthopappus* genus is a perennial herb that is endemic to the Taihang Mountains. It is listed as a second-level plant in the National Key Protected Wild Plants of China [23,24] and encompasses two species (*Opisthopappus taihangensis* and *O. longilobus*). *O. taihangensis* is located mainly in the southern Taihang Mountains of Shanxi and Henan Provinces, while *O. longilobus* distributes primarily in the northern Taihang Mountains, spanning Hebei and Shanxi Provinces. Both *O. longilobus* and *O. taihangensis* have been found growing on cliffs, being a typical lithophyte [19,24].

With the enhanced Asian monsoon from the QTP uplift, the established differing monsoon regimes resulted in the derivation of *O. taihangensis* from *O. longilobus* at ~ 17.44 Ma during the early Miocene [19]. Subsequently, the continual uplift of the Taihang Mountains intensified the differentiation between the two species. *O. taihangensis* is consequently regarded as a descendant of *O. longilobus* [19]. Based on our observations, the flowering of the two species did not completely overlap, although flowering for both proceeds from July to October. What does this non-overlap of the flowering period indicate? What differentiation exists between the reproductive characteristics of these two species? Being a descendant [19], might *O. taihangensis* possess a better adaptation than its ancestor to the cliff ecosystem due to the breeding system?

In order to address these issues; the present study was initiated to explore and gather data on *O. longilobus* and *O. taihangensis* with respect to the abovementioned parameters to understand their various phenophases toward the elucidation of potential deficiencies in their reproduction and pollination mechanisms. Specifically, we focused on the following questions: (1) What are the reproductive characteristics of these two *Opisthopappus* species? (2) How do floral traits associate with pollinators and influence their pollination success? What is the main pollination system for these species? (3) How does the breeding system of these *Opisthopappus* species promote their reproductive success? And how does the breeding system drive the evolution of these two cliff species?

The results of this study elucidate the impacts of different biotic and abiotic constraints on the phenology and reproductive biology of *O. longilobus* and *O. taihangensis*, including the unique strategies they employ to facilitate pollination and ensure reproduction. This study not only helps to explore the underly causes for the existing status of these cliff species but also assists in unraveling the critical events that occur during their life cycles.

## 2. Results

### 2.1. Flowering Phenology and Floral Traits

The *Opisthopappus* genus comprises autumn flowering species. During September, the flower buds of *O. longilobus* and *O. taihangensis* begin to develop and differentiate. The flowers bloom from mid-September to late October and peak in early October; the seeds mature at the end of November.

The flowering time of the two *Opisthopappus* species was ~60 days, and a single flower would unfold for ~seven days. Interestingly, purple or green flower buds were observed for both *O. taihangensis* and *O. longilobus*. For the green buds, the ligules of *O. taihangensis* and *O. longilobus* were always white from unfolding to end. For the purple buds, the ligules of *O. taihangensis* were general purple or lilac at first and gradually changed to light pink or white until fully unfolding, while the ligules of *O. longilobus* were pale pink or white with unfolding (Appendix A).

The floral phenology of a single capitulum could be divided into nine stages: (1) bud stage, (2) pre-bud break, (3) late bud break (bud breaks and the internal structure of the flower not present), (4) initial unfolding stage (ligules initially unfold and the internal structure begins to present), (5) ligules unfolding stage (ligules fully unfolding and florets fully exhibiting), (6) florets begin to unfold (the outermost layer of the disc florets begin to unfold, pistils mature, and stamens begin to disperse pollen), (7) florets fully unfolding (the innermost layer of the disc florets completely unfolding), (8) floral defeat (the ray ligules and disc florets begin to wither), (9) complete floral defeat (the flowers completely wither), and (10) seed maturity (seeds maturation).

Based on field observations, the developmental stages of capitulum maturation of *O. longilobus* and *O. taihangensis* were really the same without obvious differences (Figure 1A,B).

The capitulum dimensions of *O. taihangensis* were slightly larger than those of *O. longilobus* (*p* < 0.05). The numbers of ray ligules and disc florets for *O. taihangensis* were more than those of *O. longilobus* (*p* < 0.05). These showed that *O. taihangensis* had a relatively larger capitulum and more numbers of ligules and florets. The detailed floral morphological characteristics are presented in Table 1.

The capitulum of both *O. longilobus* and *O. taihangensis* were comprised of ray ligules and disc florets, where the external ligules were all female with stigmas, while the internal florets were all bisexual flowers that simultaneously contained stamens and pistils (Appendix A). In addition, it was found that most *O. longilobus* and *O. taihangensis* stigmas were two-lobed, while there were only a few that were three-lobed (Appendix A). Thus, the floral organ structures were not significantly differentiated between the two species.

All the florets of *O. longilobus* and *O. taihangensis* were hermaphrodites. The corolla of the florets was sympetalous with a five-lobed apex. The filaments of stamen attached to the edge of the corolla, as well as the anthers, were connected to form a five-lobed tube around the style. According to the stamens of *O. longilobus* and *O. taihangensis,* both were polygamous.

At the onset of flowering, both the stamens and pistils were enclosed within the petals of the florets, which opened upon blooming to expose the filaments with anthers that were gradually raised, whereafter the pollen sacs began to rupture and disperse pollen. Meanwhile, the stigmas began to elongate and become longer than the stamens, and steadily the penniform spread out (Appendix A). As relates to the internal florets, the external ones initially pinnated and dispersed pollen, subsequently followed by the internal florets (Appendix A).

The *O. longilobus* and *O. taihangensis* stigmas were initially short and rod-shaped, which soon transformed to a “Y”-shape over time and finally to a curly and oxhorn-shape (Figure 2). Scanning electron microscopy revealed that the stigmas were composed of finger-like cells at the top and protruding cells on both sides, which could secrete mucus and adhere to the pollen. Thus, the pollen was successfully deposited on the finger-like cells and protruding cells of the stigma and sprouted pollen tubes (Figure 2).

### 2.2. Floral Visitors

The behaviors of potential pollinators were significantly influenced by external environmental factors (e.g., temperature) [25]. Field observations revealed that the flower-visiting behaviors of insects occurred mostly on sunny days and showed a single-peaked curve of the daily activity pattern.

The activities of flower-visiting insects occurred mainly from 11:00–14:00, with most insects visiting from 12:00 to 13:00. On sunny days, visiting insects were rarely observed when the temperature was lower (8:00–11:00). Thereafter, the number of visiting insects decreased rapidly due to lower temperatures. Almost no flower-visiting insects were observed after 15:00 on sunny days or on cloudy/rainy days. According to comma observations, the number of *O. taihangensis* flower-visiting insects was higher than that for *O. longilobus* during the times described above (Appendix A).

A total of 14 species of insects were identified that visited the flowers of *O. longilobus* and *O. taihangensis* (Figure 3), which belonged to eight families of five orders, Namely, Vespidae (Hymenoptera), Formicidae (Hymenoptera), Syrphidae (Diptera), Cordyluridae (Diptera), Lycaenidae (Lepidoptera), Nymphalidae (Lepidoptera), Cetoniidae (Coleoptera), and the Pentatomidae stinkbug (Rhynchota).

Among these insects, *Episyrphus balteatus* (Diptera) was the most frequent flower visitor, followed by Vespula vulgaris (Hymenoptera), *Ischiodon scutellaris* Fabricius (Diptera), *Eristalis tenax* (Diptera), *Eristalis arvorum* (Diptera), and *Vespula flaviceps* (Hymenoptera) (Appendix A). These insects visited *O. taihangensis* flowers more than those of *O. longilobus*. These potentially effective pollinators have been observed alighting on the corolla of florets during whole flower opening, where they picked up pollen grains with their mouth and body parts, transferring pollen from one flower to another.

Further, *Scathophaga stercoraria* (Diptera), *Oxycetonia jucunda* Faldermann (Coleoptera), *Vespa velutina* (Hymenoptera), and the Parasitoid wasp (Hymenoptera) typically visited *O. longilobus* and *O. taihangensis* flowers. Diptera and Hymenoptera, which may also be effective pollinators, were observed landing on the corolla; however, their visiting frequency was relatively low. *Oxycetonia jucunda* Faldermann (an insect of Cetoniidae (Coleoptera)), which also alighted on *O. longilobus* and *O. taihangensis* flowers, albeit with the least visiting frequency and remaining on flowers for only a few seconds, may be a pollen thief with its proboscis.

In addition, *Vespa velutina* (Hymenoptera), *Parasitoid wasp* (Hymenoptera), and *Halyomorpha halys* (Rhynchota) mainly visited *O. taihangensis* flowers, while *Aricia mandschurica* Staudinger (Lepidoptera), and *Polygonia caureum* (Lepidoptera) primarily visited *O. longilobus* flowers (Appendix A). *Halyomorpha halys* was an occasional pollinator of *O. taihangensis* that usually landed on the petals and seldom touched the tubular flowers (Appendix A).

### 2.3. Pollen Viability and Stigma Receptivity

A pollen viability test was conducted at different flowering stages. On the first, second, and third days of anthesis, the pollen of *O. taihangensis* had the highest viability with a mean value of 92.12 ± 0.12% (Table 2). On the fourth day, the pollen viability began to decrease to 80.4 ± 0.47% and continually decreased on the fifth, sixth, and seventh days.

The pollen viability of *O. longilobus* exhibited a sequence similar to that of *O. taihangensis* (Appendix A), with the highest viability also occurring during the first three days, at 88.69 ± 0.09%, 86.22 ± 0.25%, and 81.76 ± 0.17%, respectively. Beyond the fourth day, the viability of *O. longilobus* pollen began to decrease. On the fourth, fifth, sixth, and seventh days, the viability was 68.52 ± 0.85%, 52.90 ± 0.33%, 45.31 ± 0.39%, and 39.44 ± 0.15%, respectively (Table 2).

These data suggested at least a seven-day lifespan for the pollen grains of *O. taihangensis* and *O. longilobus*. Meanwhile, the pollen viability of *O. taihangensis* was superior to that of *O. longilobus* (*p* < 0.05).

The change trends for *O. taihangensis* and *O. longilobus* pollen viability were similar on the same flower unfolding day, as they both displayed a single peak (Table 2). The pollen viability of the two species gradually increased from 8:00 to 13:00 and then decreased from 13:00 to 17:00. The highest pollen viability was found at 13:00 for both *O. taihangensis* and *O. longilobus*. Moreover, the viable pollen number of *O. taihangensis* was higher than that of *O. longilobus* during the same times (*p* < 0.05).

The receptivity of the stigmas was found to gradually increase during the first 1–4 days of the flowering period (Table 3, Figure 4). For *O. taihangensis*, they exhibited the highest receptivity on the third and fourth days, while the highest level of *O. longilobus* stigma receptivity occurred on the fourth day. Subsequently, the stigma receptivity for the two species began to decrease on the fifth and sixth days, then decreased to zero on the seventh day (Table 3, Figure 4).

On the same day, during the flowering phase, the stigma receptivity of *O. taihangensis* and *O. longilobus* exhibited a similar change pattern (Table 3, Figure 4). The stigmas of the two species showed the highest receptivity level at 11:00–13:00, around which abundant bubbles were emitted. Subsequently, the stigma receptivity decreased with fewer bubbles over time.

At a different stage, it was observed that the closed stigmas with few bubbles exhibited weak receptivity. When the stigmas opened and appeared in their “Y” configuration, their receptivity increased around more bubbles. When the stigmas appeared as rolled Y shape, the receptivity displayed the highest level around ample bubbles (Figure 4).

### 2.4. Outcrossing Index and Pollen/Ovule Ratio

The average diameters of the floret’s inflorescence (bearing part of the capitulum) of *O. taihangensis* and *O. longilobus* were 5.68 ± 0.12 cm and 4.55 ± 1.22 cm, respectively. Both of these diameters were longer than 0.6 cm and thus were scored as 3 according to the Dafni criterion [26]. When the florets of *O. taihangensis* and *O. longilobus* unfolded, the anthers in the same capitulum could dehisce and disperse pollen during the stigma receptivity period, although having protandry. This indicated that completely self-reproductive isolation did not occur in *O. taihangensis* or *O. longilobus*. Rather, a degree of self-compatibility occurred in the two species; thus, they were scored as 0. Meanwhile, the stigmas were longer than the anthers for *O. taihangensis* and *O. longilobus* (*p* < 0.05), which formed a certain degree of spatial isolation with a score of 1.

According to Dafni’s criterion, the outcrossing index (OCI) for both *O. taihangensis* and *O. longilobus* was considered to be 4. This revealed that the reproductive system should be an outcrossing model with partial self-compatibility, which required pollinators to transfer pollen during the breeding process.

For each floret of *O. taihangensis*, the average pollen grain was 1928 ± 236, while the average pollen grain in each floret of *O. longilobus* was 1321 ± 309. Only one ovule was observed for both *O. taihangensis* and *O. longilobus* of per flower. Thus, the pollen/ovule ratio (P/O) of a single flower for the two species was 1928 and 1321, respectively. Based on the Cruden criterion [27], the breeding system of both species was facultative xenogamy, which was consistent with the OCI classification.

### 2.5. Seed Set and Germination Rates

Without bagging and artificial pollination, the seed rates were 40.87% and 37.66% for *O. taihangensis* and *O. longilobus*, respectively. If bagging occurred prior to pollination scattering, the seed set rate of *O. taihangensis* was 23.23%, while that of *O. longilobus* was 20.06%. The 30.28% seed set rate for *O. taihangensis* and 26.06% for *O. longilobus* was observed under the geitonogamous pollination treatment. Under the xenogamous pollination treatment, the seed set rate was 35.71% for *O. taihangensis* and 28.57% for *O. longilobus*.

Under different treatments, the seed set rate values generally exceeded 20%, which further suggested that *O. taihangensis* and *O. longilobus* were self-compatible to a certain extent.

According to the above different treatments, the mature seeds were respectively collected at different sites from 2018 to 2019, and seed germination was subsequently performed at room temperature conditions in the laboratory. Then, we found that the average seed germination rates were 53.33 ± 2.01% for *O. taihangensis* and 43.33 ±1.47% for *O. longilobus*. Meanwhile, the seed germination potentials of *O. taihangensis* and *O. longilobus* were 36.00 ± 0.28% and 25.00 ± 1.02%, respectively.

According to the above, the seed germination rate and potential of *O. taihangensis* were higher than that of *O. longilobus* (*p*< 0.05), even at the same sampling site.

## 3. Discussion

### 3.1. Floral Syndromes and Effective Pollination

Floral syndromes comprise sets of integrated floral traits (e.g., morphology, color, and scent) [4,25,28]. For the two species of *Opisthopappus*, the floral syndromes were similar (Figure 1) except for the capitulum size and the number of ligules and florets (Table 1). As the descendant of *O. longilobus* [19], *O. taihangensis* presented floral syndromes that were similar to those of its ancestors. On the other hand, the relatively complicated floral syndromes of *O. taihangensis* might be an evolutionary outcome. With a larger capitulum, more ligules, and florets, *O. taihangensis* could better attract pollinators for reproductive success, development, and survival when it faces different habitats compared with *O. longilobus*.

For plants, floral traits are associated with different pollination syndromes [4], which may evolve to align with the morphologies and behaviors of pollinators [29]. The floral syndromes of *O. taihangensis* and *O. longilobus* (e.g., bisexual disc florets, brightly white ligules, brightly yellow florets, and protandry) indicated that their pollination systems should be animal-mediated cross-pollination. In the lower portion of the florets, nectary secretory tissue lined and generated a certain amount of nectar. Nectar can release a fragrance and provide precious forage for flower-visiting insects, particularly in mid-autumn. The capitula inflorescences with receptacles are beneficial for insect visitors to remain for prolonged periods. Thus, the results of flower-visiting observation pollination support that the pollination system of *O. taihangensis* and *O. longilobus* should be animal-mediated cross-pollination.

The main floral visitors of *O. taihangensis* and *O. longilobus* were found to be hymenopteran and Diptera species (Figure 3). Our results confirmed that they shared the same flower-visiting insects, as they possess similar floral morphologies (Appendix A). Meanwhile, the flowers have characteristics that are associated with bee and fly pollination (Figure 3), such as small tubular corollas and sites with concentrated nectar. Bees generally have pollen-carrying legs that are completely brosse [7,30]. Visiting bees can easily carry pollen when landing on the florets and transport it from one flower to the next. Since the bodies of flies are generally smaller than those bees, they may enter the corolla tube to feed on nectar and then repeatedly contact the stigma of different flowers during foraging [30,31]. Thus, the data suggested that bees and flies (e.g., *Vespula vulgaris* and *Episyrphus balteatus*) were the effective pollinators of *O. taihangensis* and *O. longilobus* (Figure 3, Appendix A).

Interestingly, *O. taihangensis* and *O. longilobus* were observed to have unique flower-visiting insects (Appendix A). Two Hymenoptera species (*Vespa velutina* and *Parasitoid wasp*) were specific for *O. taihangensis*, which can not only land on but also enter the florets with their slender bodies. Meanwhile, two Lepidoptera species, *Aricia mandschurica* Staudinger and *Polygonia caureum* were specific for *O. longilobus* (Figure 3). The butterflies have siphoning mouthparts and can suck nectar through their long tubular beaks. When they visit flowers, their body hairs can adhere to and transfer pollen [31,32]. The differentiation of the flower-visiting insects between *O. taihangensis* and *O. longilobus* may reflect the floral syndrome variations, such as the flower size and color, particularly the flower fragrances, where different aromas might attract various insects. Metabolome data revealed that significantly different floral metabolites were found between *O. taihangensis* and *O. longilobus* [33].

The extended floral longevity of these plants may be an evolutionary strategy that is employed to prevail over sparse or unpredictable pollinator services by increasing the quantities of their own exported pollen and imported foreign pollen [4,34]. Our results revealed that the flowering duration of the two *Opisthopappus* species extended for about 60 days, while the anthesis of a single flower continued for six days. The pollen release time was relatively concentrated. The pollen of *O. taihangensis* and *O. longilobus* gradually matured from the external to internal florets (Table 2, Appendix A), with blossoms being relatively concentrated in October. The extended flowering time and pollen presentation could increase the time of pollen collection for pollinators [35]. Furthermore, the extended floral longevity was likely an adaptation for survival and development in the difficult cliff environment of the Taihang Mountains.

For most flowering plants, pollen viability and stigma receptivity are critical parameters for the successful initiation of pollen-pistil interactions [7,36]. Extended stigma receptivity assures the pollination, fertilization, and reproductive success of plants [14]. The stigma of *O. taihangensis* and *O. longilobus* can remain receptive for seven days to provide the proper conditions for pollen to germinate and affect fertilization (Table 3, Figure 4). This time frame appeared to be a strategy for ensuring the reproductive output of the two species [7]. The anther dehiscence in *O. taihangensis* and *O. longilobus* occurred simultaneously with anthesis. Alternate anther dehiscence appeared to be an adaptation for efficient and extended pollen availability to the pistil, which may further increase the chances of being pollinated [4,7].

Despite successful insect pollination, the activities of pollinators can be easily influenced by weather conditions, as in this study, almost no visiting insects were observed on rainy or cloudy days. We found that the flowering periods of the two *Opisthopappus* species of sampled sites basically overlapped with the rainy season (November to October), which had a negative impact on the flower-visiting behaviors of insects. Under this situation, successful reproduction may rely solely on self-pollination or geitonogamy [18].

Insect pollinators preferred warmer noon hours to visit the flowers, after which their visitation frequency decreased (Appendix A) [7,37]. Different plant species experienced a drastic decrease in the visitation efficiency of pollinators during windy and cloudy periods and rain [38]. Adverse climatic conditions were found to hamper the visitation frequency and efficacy of the pollinators [18]. Lower anthesis temperatures were unfavorable for the pollinators [7]. The low temperatures of early autumn in North China may also decrease the diversity and activities of pollinators [39], especially in the cliff habitats of the Taihang Mountains [4,40]. However, the long anthesis period of each flower, as well as the capitulum and light flower color, with shared pollinators, still improved the pollination efficiency and resulted in the mass production of seeds for *O. taihangensis* and *O. longilobus*.

### 3.2. Breeding System

The breeding systems of plants are a manifestation of the interactions between their internal genetic mechanisms and the external environment, which plays an important role in the evolution and variation of species [41]. Knowledge regarding the breeding system of a species is beneficial for elucidating the characteristics of evolution and life history caused by different genetic and ecological factors [6].

It is widely known that the outcrossed progeny of organisms are expected to be more heterozygous; hence, individually more adaptive and with more fitness than selfed progeny [18,42]. However, in flowering plants, pure outcrossing (e.g., self-incompatible) may bring potential risks to the number of offspring, as cross-fertilization relies on an external agent for pollen transfer (e.g., animals and wind), which is often unpredictable and/or inadequate [6]. Thus, in nature, there are very few plants that are exclusively self-crossing or out-crossing. Instead, most modern plant species evolved toward a mixed mating system combining self-fertilization and cross-fertilization for reproductive success [4,41], particularly as a strategy to cope with unstable environments [4,43].

According to Dafni [26], the out-crossing index (OCI) was 4 for this study, which indicated that these *Opisthopappus* species were self-compatible. Further, the pollen-ovule ratio (P/O) was in the range of 244.7–2588.0, which corresponded to the values reported for facultative xenogamy [43]. Hence, *O. taihangensis* and *O. longilobus* were a mixture of self-pollination and out-crossing. The mixed breeding strategy ensures reproductive output that not only ensures sexual reproduction but also generates variation and distributes effectively [7].

The evolution of plant mating strategies always has one main theme related to the evolution of cross- vs. self-fertilization [44,45]. Predominant selfing and outcrossing should be alternative stable outcomes of mating system evolution for most plants [25,46]. Our results revealed that *O. taihangensis* and *O. longilobus* could propagate via simultaneous selfing and outcrossing; however, outcrossing predominated. The seed set from geitonogamous, xenogamy, and control was higher, respectively, than that from autogamy. The seed germination rate from autogamy was lower than that from geitonogamous, xenogamy, and control, which further supported that outcrossing prevailed in the breeding system. Self-pollination played a secondary role in ensuring production in the breeding system when the conditions for outcrossing were unfavorable due to shifting environment conditions, or in cases when normal pollinator populations were reduced or absent [25].

Dichogamy and herkogamy are the two main mechanisms that prevent spontaneous self-fertilization [4,47]. Although *O. taihangensis* and *O. longilobus* both displayed protandry, the seed rate of manual self-pollination exceeded 20%, which indicated that spontaneous selfing occurred. These two species exhibited several floral traits that were specifically associated with self-pollination (bisexual florets and no temporal separation between pollen viability and stigma receptivity) [48]. However, the fruit set and germination rate from autogamy were significantly lower than that from xenogamy and control.

Inbreeding/selfing is relatively common in species with small populations and individuals, narrow habitats, or modest founding populations [6,49,50] and is of great significance for their preservation and reproduction. Both *O. taihangensis* and *O. longilobus* grow on cliffs of the Taihang Mountains and thus are subjected to a harsh environment. Sustained self-compatibility would be a strategy during the evolutionary process, as a delayed mechanism for reproductive assurance [51,52], or even as a main contributor to natural seed setting [18].

Moreover, selfing results from the activities of pollinators. The numbers of flowers unfolding simultaneously are important for the reproductive success of species that are dependent on biotic vectors since this can increase pollinators’ attraction and visitation [53,54]. However, visits will frequently result in geitonogamous self-pollination [4,55]. Thus, the high fruit set rate under ambient conditions for *O. taihangensis* and *O. longilobus* is likely the result of both cross-pollination and geitonogamy.

## 4. Materials and Methods

### 4.1. Study Species and Sites

*O. longilobus* and *O. taihangensis* are autumn flowering plants which possess pinnatifid leaves and heterogamous capitulum. Their ray ligules are general lamina linear, white, or pink. The disk florets are bisexual, yellow, and five-lobed. The fruits and seeds are a type of obovoid achene, 1.2 mm in length, with three or four ribs [56].

*O. longilobus* and *O. taihangensis* typically grow on steep cliffs at altitudes of from 1000- 2000 m, along with other coexisting plants (some Gramineae) in wild habitats. Due to the harsh environment, they were difficult for us to observe. Thus, we created a common garden in Linfen City (CLF) of Shanxi Province, along with transplant gardens at Shennongshan (SNS) in Henan Province and Xiangtangshan (XTS) of Hebei Province. Our fieldwork was conducted at these three sites from July to November.

The CLF site is located at the Shanxi Normal University Nursery (36°60′ N, 111°30′ E) in Shanxi, China. The climate of the CLF belongs to a type of temperate continental monsoon, which is characterized by cold winters with little snow, spring winds, autumn rains, and hot summers with droughts. The average annual temperature is 12.6 °C, with 500–600 mm of precipitation and a frost-free period of 190 days [57]. The SNS is set in the Shennong Mountains (35°11′30″ N–35°19′ N, 112°44′ E–113°02′ E) in Henan, China. The climate here is warm-temperate continental with four distinct seasons, which are dry and windy in spring, hot and rainy in summer, with warm days and cool nights in the autumn, and a cold and dry winter. The average annual temperature is 14.3 °C, with maximum/minimum temperatures of 42.1 °C and −18.6 °C, respectively [58]. The XTS is located in the Xiangtang Mountains (36°20′ N–36°20′ N, 114°03′ E–114°16′ E), in Hebei, China, with a warm temperate continental monsoon climate. This site also has four distinct seasons, with an average annual temperature of 13.5 °C and an annual frost-free period of 200 days [59].

### 4.2. Observation of Flowering Phenology and Floral Traits

To determine the flowering patterns of *O. longilobus* and *O. taihangensis*, 30 capitula of each species were randomly selected for observation. The different flower stages (e.g., buds, ligules unfolding, and flowering) of *O. longilobus* and *O. taihangensis* were observed and recorded using a Sony (DSC-W830) camera. Each flower was labeled with tying on the label during the sympetalous period and observed every day until unfolding.

Subsequently, 60 flowers (three flowers per individual) in full bloom were randomly selected to measure and record the attributes of the ligules and florets under a double-tube dissecting microscope (PXS-1014). The capitulum diameter, the diameter of all florets, ligule length, ligule width, number of ligule flowers, and number of florets of *O. taihangensis* and *O. longilobus* were also measured and recorded (60 replicates for each species).

Additionally, the features of stigma at different flowering stages were initially observed by optical microscopy (PXS-1014) and then photographed by scanning electron microscopy. For the florets, the capitulum of *O. taihangensis* and *O. longilobus* was observed via optical microscopy (PXS-1014) to analyze the unfolding order of the florets on the capitulum. The characteristics of the florets at different flowering stages were also observed and photographed via optical microscopy (PXS-1014).

### 4.3. Pollinator Observations and Pollination Experiments

Insect observations were conducted continuously during peak flowering in the field, from 10:00 to 18:00 for 60 d in 2018 and 2019, from September to October. Each insect was placed in a separate vial, where the presence or absence of pollen grains adhered to their bodies was determined in the laboratory using a stereomicroscope. The species were identified by associate professor Junxiang Cai (an insectologist at Shanxi Normal University). The insect specimens were preserved in the specimen room of the School of Life Science at Shanxi Normal University.

The times and frequencies of insect visitors to *O. longilobus* and *O. taihangensis* flowers were recorded over 30 consecutive days, which totaled 500 flowers under observation for each species. The pollinators and their foraging behaviors were analyzed by direct visual observations, which were complemented by photographs taken during visits to the three sites. Based on their visitation frequency and behavior on the flowers, the insects were classified as effective pollinators or occasional pollinators. Effective pollinators represented the frequent insects (with a higher number of visits) that carried pollen grains from one anther of a flower to the stigma of another. Occasional pollinators represented those insects that showed the same behavior but with a low visitation frequency. Insects that only collected pollen and/or nectar without pollinating the flowers were classified as thieves [60,61].

### 4.4. Pollen Viability and Stigma Receptivity

After flowering, ten flowers were sampled daily at 1, 2, 3, 4, 5, 6, and 7 days. Seventy flowers were randomly selected and remarked for *O. longilobus* and *O. taihangensis* receptively. The pollen was sampled every hour from 8:00–17:00 for seven consecutive days, with three replicates for each sample.

The pollen viability and stigma receptivity were detected by the I_2_-KI method at different developmental stages. Some of the pollen suspension was placed on a microscope slide using a pipette, and the pollen grains were observed using an optical microscope (10 × 40x, OLYMPUS BX51 EVOS-M7000). Five fields were randomly selected from each slide to observe the staining status of the pollen grains. The deeply/completely stained pollen grains were considered viable, according to Dafni [26].

The stigma receptivity was analyzed using the benzidine-hydrogen peroxide technique [61], wherein bubbling in the presence of hydrogen peroxide was considered to be a positive result [62,63]. The reaction was observed and recorded under an optical microscope (10 × 10x). We also recorded the time when bubbles appeared on the stigma, where their abundance around the stigma indicated viability. The absence of bubbles around the stigma indicated inactivity or sterility. The stigma receptivity was verified using the catalase activity method [25].

### 4.5. Outcrossing Index and Pollen/Ovule Ratio

The outcrossing index (OCI) and ratio of pollen grains to ovules (P/O) were determined to estimate the likelihood of pollination outcrossing (xenogamy) versus selfing (autogamy) based on the floral morphological characteristics [4,44,62].

For the pollen count, the anthers were collected from the synantherous stamen of mature flowers, from which three florets were collected. A total of 30 florets from ten randomly selected *O. longilobus* and *O. taihangensis* were examined. The synantherous stamen was opened with fine-tipped forceps, and the anthers of each flower were transferred to a centrifuge tube (1.5 mL) filled with 1 mL distilled water after drying. Next, a 1 μL subsample of the pollen suspension was transferred to a microscope slide using a pipette, and the pollen grains were counted using an optical stereomicroscope (10 × 10x, EVOS-M7000).

The total number of pollen grains = subsample a number of pollen grains × volume of centrifuge tube/volume of pollen suspension in the pipette. This measurement was repeated 10 times for each flower, after which the maximum and minimum values were removed before averaging. The total pollen number for each flower was recorded as P, and the number of ovules of each flower was counted and recorded as O. The pollen-ovule ratio was calculated following the method of Cruden [27].

The OCI was estimated as the sum of values from: (1) flower diameter, 0, 1, 2, 3 for opening diameters of ≤1, 1–2, 2–6, > 6 mm, respectively; (2) spatial position between anther and stigma, 0 remarked at the same height and 1 for separation from each other; (3) interval time of anther dehiscence and when the stigma was presently fertile, 0 for anther and stigma both maturing at the same time or the gynoecia initially maturing, and 1 for the stamen initially maturing [44,62]. Namely, the sum of (1) + (2) + (3) was the OCI value.

When OCI was 0, the breeding system of the species was designated as closed-fertilized; when OCI = 1, the breeding system was designated as exclusively self-fertilized; when OCI = 2, it was as a parthenogenetic system for the species; when OCI = 3, the system was considered as a self-fertilized type that sometimes required pollinators; when OCI ≥ 4, the breeding system was considered as partially self-fertilized with a heterozygote pattern that required pollinators.

### 4.6. Seed and Germination Rates

To better characterize the mating pattern of *O. longilobus* and *O. taihangensis*, four treatments were established for the flowers: (1) control (freely pollinated flowers without any treatment), (2) freely pollinated (bagged before flowering with no emasculation), (3) geitonogamous pollinated (emasculated and bagged before pollen scatter of the florets, artificially pollinated between different flowers on the same individual), (4) artificially pollinated (bagged and outcrossed with pollen from other individuals). Each treatment involved 15 individuals. Within each individual, three capitula were randomly selected. When the seeds matured, those under each treatment were collected.

The seeds collected from each treatment after they matured from October to November from 2018 to 2019. Under the room temperature of the library, the germination experiments of the seeds were carried out. Fifty randomly selected seeds of each treatment with three replicates were placed in one culture dish for each species. Subsequently, water was regularly and quantitatively supplied for all culture dishes, and the germination changes of the seeds were recorded every day.

The germination rate was the percentage of germinated seeds to the number of seeds in the experiment, whereas the germination potential was the percentage of germinated seeds to the number of seeds in the experiment on the sixth day.

### 4.7. Statistical Analyses

Datasets with a normal distribution were analyzed by Student’s *t*-test and one-way ANOVA. All analyses were performed in SPSS 20.0 (SPSS, Chicago, IL, USA) for Windows. All parameter estimates were calculated as mean ± SE (Standard Error) values. The data were plotted using Origin 2.0 software.

## 5. Conclusions

Reproductive biology is an important interdisciplinary area of plant sciences, which is essential for understanding the evolution and survival of species. For this study, we investigated the floral biology, breeding systems, and pollination ecology of two *Opisthopappus* species (*O. taihangensis* and *O. longilobus*). These plants exhibited a suite of floral syndromes (e.g., bisexual florets, bright white ligules, bright yellow florets, big capitulum, and distinctive fragrances) that adapted to insect-mediated cross-pollination. Comprehensive studies of the OCI, P/O ratio, and artificial pollination suggested that *O. taihangensis* and *O. longilobus* had a highly pollinator-dependent mixed mating system of cross- and self-fertilization. Outcrossing predominated in the mating system, and autonomous self-pollination in its breeding system may have provided reproductive assurance for the sustainability of the population. This allowed for successful reproduction in the harsh cliff environment of the Taihang Mountains. The reasonable number of seeds produced indicated that the species were capable of sustaining their progenies in natural populations. Although bees and flies were their effective pollinators, *O. taihangensis* and *O. longilobus* each possessed unique pollinators. As the descendant of *O. longilobus*, *O. taihangensis* with special pollinators and relatively complex floral syndromes (e.g., pink and white ligules, more numbers of ligules, and florets) exhibited improved adaptation over its ancestor. Our results not only highlighted the reproductive characteristics of an endemic species of the Taihang Mountains of North China but also provided insights into the reproductive characteristics of species that flourish in limestone mountain habitats.

## Figures and Tables

**Figure 1 plants-12-01954-f001:**
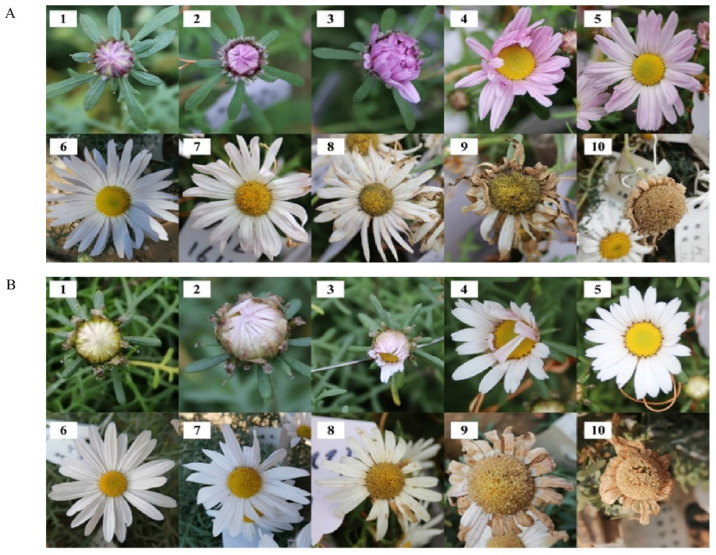
Developmental stages of capitulum maturation of *Opisthopappus taihangensis* and *Opisthopappus longilobus* (1 bud stage; 2 pre-bud break stage; 3 bud break stage, 4 initial unfolding stage; 5 ray ligules unfolding stage; 6 florets unfolding stage; 7 florets full unfolding stage; 8 beginning wither stage; 9 fully wither stage; 10 seed maturation stage). (**A**): Developmental stages of capitulum maturation of *O. taihangensis*; (**B**): Developmental stages of capitulum maturation of *O. longilobus*.

**Figure 2 plants-12-01954-f002:**
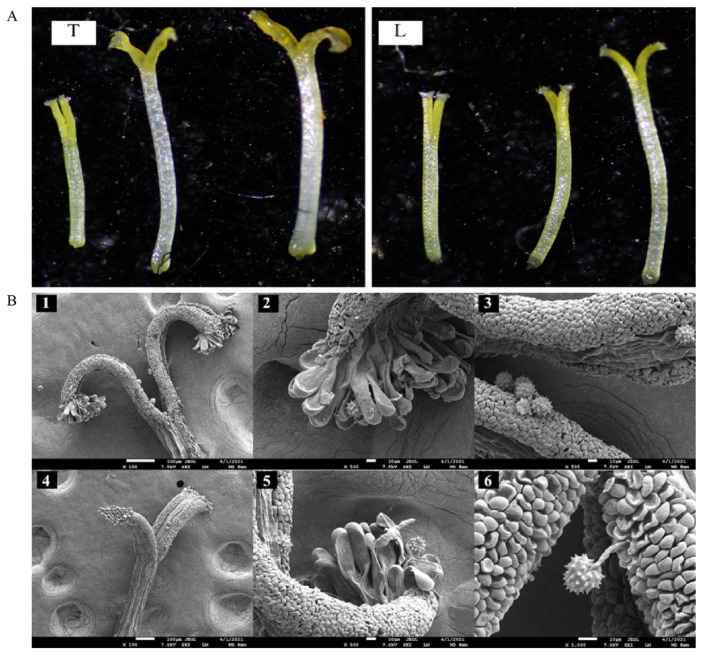
Stigmas of *Opisthopappus taihangensis* and *O. longilobus* (T: *O. taihangensis*; L: *O. longilobus*).(**A**): dynamic changes (5×); (**B**): Scanning electron microscopy (1 stigmas of *O. taihangensis*; 2 implantation and germination of *O. taihangensis* pollen on stigma finger cells; 3 implantation and germination of *O. taihangensis* pollen on protuberant cells; 4 stigmas of *O. longilobus*; 5 implantation and germination of *O. longilobus* pollen on stigma finger cells; 6 implantation and germination of *O. longilobus* pollen on protuberant cells).

**Figure 3 plants-12-01954-f003:**
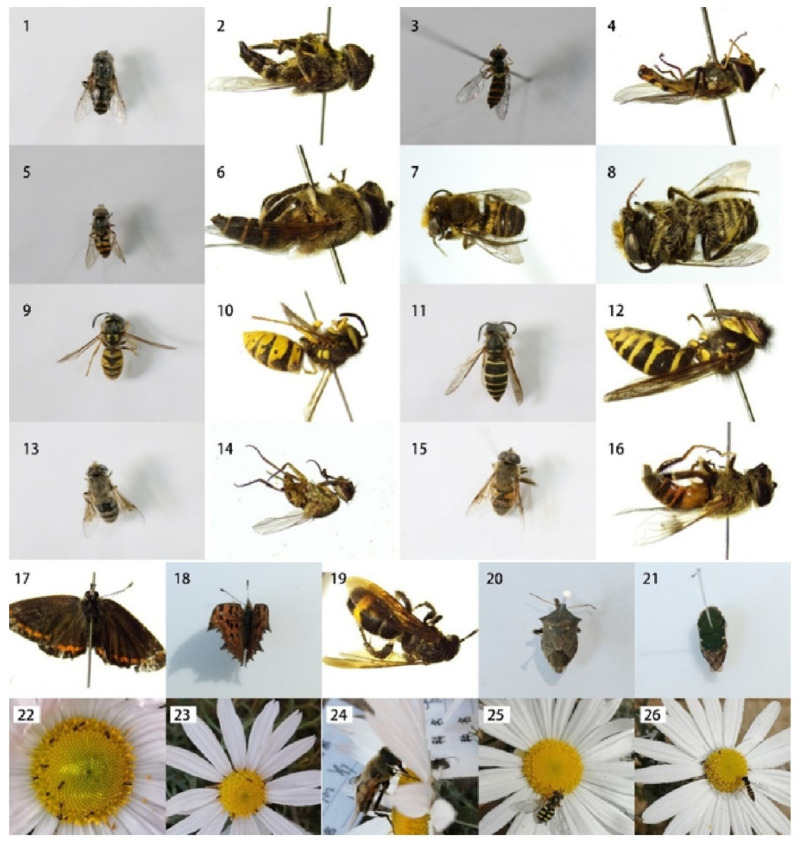
Visiting flower insects and their activities (1–2) *Eristalis arvorum*, (3–4) *Ischiodon scutellaris* Fabricius, (5–6) *Episyrphus balteatus* De Geer, (7–8) *Parasitoid wasp*, (9–10) *Vespula vulgaris*, (11–12) *Vespula flaviceps*, (13–14) *Scathophaga stercoraria*, (15–16) *Eristalis tenax*, (17) *Aricia mandschurica* Staudinger, (18) *Polygonia caureum*, (19) *Vespa velutina*, (20) *Halyomorpha halys*, (21) *Oxycetonia jucunda* Faldermann, (22) *Pheidole megacephala* visiting *O. taihangensis*, (23) *Pheidole megacephala* visiting *O. taihangensis*, (24) *Eristalistenax* visiting *O. taihangensis*, (25) *Episyrphu sbalteatus* De Geer visiting *O. taihangensis*, (26) *Ischiodon scutellaris* Fabricius visiting *O. longilobus*.

**Figure 4 plants-12-01954-f004:**
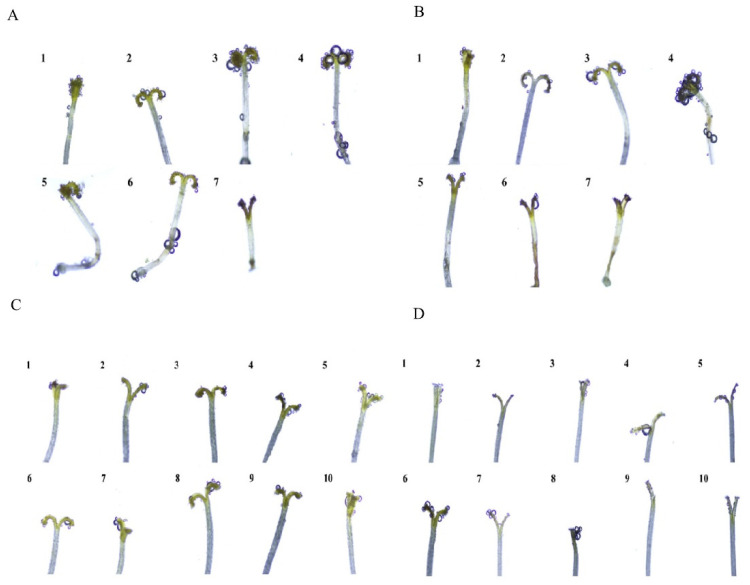
Stigma acceptability of *Opisthopappus taihangensis* and *Opisthopappus longilobus*. (**A**): The stigma acceptability of *O. taihangensis* on different flowering days (4×) (1–7: *O. taihangensis* blooming for 1–7 days); (**B**): The stigma acceptability of *O. longilobus* on different flowering days (4×) (1–7: *O. longilobus* blooming for 1–7 days); (**C**): *O. taihangensis* stigma acceptability at different times of the same day during full flowering (4×) (1–10: *O. taihangensis* stigma acceptability at 8:00–17:00 on the same day during full flowering); (**D**): *O. longilobus* stigma acceptability at different times of the same day during full flowering (4×) (1–10: *O. longilobus* stigma acceptability at 8:00–17:00 of the same day during full flowering).

**Table 1 plants-12-01954-t001:** Flower traits of *Opisthopappus taihangensis* and *Opisthopappus longilobus* flowers.

	Capitulum Diameter (mm)	Ligules Number	Ligules Length (mm)	Ligules Width (mm)	Florets Number	Diameter of All Florets (mm)
*O. taihangensis*	47.09 ± 1.67 a	29.70 ± 2.64 a	21.50 ± 1.08 a	5.46 ± 0.67 a	302.52 ± 65.31 a	14.11 ± 0.38 a
*O. longilobus*	38.70 ± 0.85 b	23.20 ± 1.74 b	16.36 ± 1.20 a	4.99 ± 0.45 a	285.04 ± 60.27 b	12.18 ± 0.77 a

Note: the first value was the mean of flower traits; the second value was the standard deviation (SD). a, no-significant variation; b, significant variation; *p*< 0.05.

**Table 2 plants-12-01954-t002:** Pollen viability of *Opisthopappus taihangensis* and *Opisthopappus longilobus* on different flowering days and different times of the same day.

	1d		2d		3d	4d	5d	6d		7d
*O. taihangensis*	93.69 ± 0.01%		91.93 ± 0.09%		90.75 ± 0.27%	80.4 ± 0.47%	60.24 ± 2.29%	53.89 ± 1.08%		40.73 ± 1.54%
*O. longilobus*	88.69 ± 0.09%		86.22 ± 0.25%		81.76 ± 0.17%	68.52 ± 0.85%	52.90 ± 0.33%	45.31 ± 0.39%		39.44 ± 0.15%
	**8:00**	**9:00**	**10:00**	**11:00**	**12:00**	**13:00**	**14:00**	**15:00**	**16:00**	**17:00**
*O. taihangensis*	70.0 ± 0.05%	83.5 ± 0.14%	88.7 ± 0.22%	90.3 ± 0.19%	96.0 ± 0.32%	97.2 ± 1.60%	89.3 ± 0.15%	83.3 ± 0.20%	80.9 ± 3.11%	76.1 ± 0.29%
*O. longilobus*	68.7 ± 0.09%	78.9 ± 1.39%	82.2 ± 0.19%	88.4 ± 0.28%	94.9 ± 0.17%	96.4 ± 0.11%	87 ± 1.03%	83.5 ± 0.03%	82.2 ± 0.03%	73.7 ± 1.81%

Note: the first value was the mean of pollen viability; the second value was the standard deviation (SD). The pollens were sampled at 8:00–17:00 for 1, 2, 3, 4, 5, 6, and 7 days.

**Table 3 plants-12-01954-t003:** Stigma receptivity of *Opisthopappus taihangensis* and *Opisthopappus longilobus* on different flowering days and different times of the same day.

	1d		2d		3d	4d	5d	6d		7d
*O. taihangensis*	+		++		+++	+++	++	+		-
*O. longilobus*	+		++		++	+++	++	+		-
	**8:00**	**9:00**	**10:00**	**11:00**	**12:00**	**13:00**	**14:00**	**15:00**	**16:00**	**17:00**
*O. taihangensis*	++	++	++	+++	+++	+++	++	++	++	++
*O. longilobus*	++	++	++	++	++	+++	++	++	++	++

Note: +, presenting the receptivity; ++, a stronger receptivity; +++, the strongest receptivity; -, a very weak receptivity or sterility. The pollens were sampled at 8:00–17:00 for 1, 2, 3, 4, 5, 6, and 7 days to observe.

## Data Availability

Not applicable.

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
