# Peer review of "Reproductive Biology and Breeding Systems of Two Opisthopappus Endemic and Endangered Species on the Taihang Mountains"

_plants, 2023, doi:10.3390/plants12101954_

Round 1

Reviewer 1 Report

The authrors present their results on the reproductive biology and breeding systems of two Opisthopappus species in China. I find the manuscript interesting, and their results are promising for publication. However, there are many problems in the manuscript, particularly with respect to the correct use of botanical terminology. In its current form, the manuscript cannot be published. The following comments may help the authors. Apart from my observations, I strongly recommend that the authors study their manuscript very carefully, offer more clarity to their sentences, make good use of the English language and be very careful with respect to the botanical terminology. The latter should apply correctly throughout the manuscript. 

line14: O. taihangensis has relatively larger, more ligulate, and tubular flowers. - I have difficulties to understand this sentence. Does it mean that O. taihangensis has larger ligules and a larger number of florets per capitulum?

Line 15: characteristics – You are referring to only one character, protandry. Please use singular.

Line 19-20: albeit certain visiting insects were also observed for the two species. – You mean insects different that Hymenoptera and Diptera or that there are some common insect species visiting the two species?

Line 25-26: Meanwhile, O. taihangensis descendants could better adapt to the severe surroundings than its ancestor O. longilobus … - Again, I feel that this sentence needs a more precise and clear meaning.

Lines 68-69: Please provide some more details about the ancestor-descendant relationships between the two species. On which knowledge / investigation do you base your statement? How is this related to the uplifting of the Taihang Mountains?

Line 87: This study not only helps to improve the existing status of these cliff species – Please consider rewording to make the meaning of the sentence clearer.

Line 91: The Opisthopappus genus are an autumn flowering species – this does not make sense. I propose: The Opisthopappus genus comprises autumn flowering species. During September …..

Line 95: the word ‘’two’’ is written twice.

Lines 97-101: the meaning is uncertain. Do you mean that the ligulate florets of O. taihangensis are pink when in bud and remain pink while unfolding, turning white the 5th-6th day? Please consider rephrasing the whole sentence.

Line 102: Instead of writing ‘’floral phenology of a single flower’’ please write ‘’ floral phenology of a single capitulum’’. The flowering head of Opisthopapus is an inflorescence (capitulum) that includes both ligulate (ray) flowers and disk (tubular) flowers. When you describe floral phenology (lines 102-110) please be very precise. Instead of using the words ‘’open, begins to open, opening’’ please use the word ‘’mature, maturation’’ etc. You can also write that the connected corolla lobes of the tubular florets split apart exhibiting reproductive organs. 

Line 114: Instead of ‘’Flower opening model’’ please write: ‘’Developmental stages of capitulum maturation’’ or something similar. Again, the whole process refers to the whole head (capitulum), not to each separate floret.

Line 120: Instead of ‘’ diameters of the capitates’’ please write ‘’the diameter of the capitula (or flowering heads)’’. Please also correct accordingly the legend of Table 1. Do the same throughout the manuscript, whenever by the word ‘’capitate’’ you mean the flowering head.

Line 121-122: Please write: The number of ligulate flowers per capitulum …..

Lines 119-127: Please add ‘’mm’’ where necessary.

Line 128: When you write ‘’flower dimensions’’ you mean capitulum dimensions?

Table 1: Tubular flower diameter (mm) cannot be 12.18±0.77! Tubular flowers are small, tiny! Please correct accordingly.

Line 140: Please be careful with the use of the word ‘’monoecious’’. It does not fit here. Please correct accordingly. Also, the term ‘’ the stamens were also five-lobed’’ does not make sense. Each stamen has an anther. The anthers are connected to form a five-lobed tube around the style.

Lines 170-173: You repeat (written twice) that peak of insect visits was between 12.00 and 13.00.

Lines 178-179: Use italics for plant names.

Line 181: Diptera is written twice.

Fig. 3. Please carefully check the spelling of all scientific names. Use authorship for all names or omit authorship for all names  (you have put authorship in some names). The same applies to the text (lines 191-210). Please remember that all scientific names are written in italics.

Table 2. Please correct: Pollen viability of Opisthopappus taihangensis and O. longilobus ….

Table 2. You counted pollen viability at different hours of the day. Which day did you make the counts? The same applies to Table 3.

Line 262: ‘’The average diameters of the tubular inflorescences’’ is not very well understood. You mean the diameter of the tubular florets -bearing part of the capitulum? Also, the meaning of lines 266-271 is not well understood. Please consider rephrasing.

Line 265: Instead of putting a date please put a reference number within brackets. The same on lines 281, 396, 508, 533.

Lines 277-279. Please avoid using the term ‘’population’’ for pollen grains.

Lines 263-269: Please consider re-wording to make meanings clearer.

Line 279: Only one ovule was observed …. Please add ‘’per flower’’.

Line 296-299 (and also paragraph 4.6). Please provide some more experiments about germination experiments. What time of the year did you make the experiments? Under which conditions?

Line 304: capitulum size, instead of flower size.

Line 306-307: Moreover, the relatively larger and more complex floral syndromes …. The wording larger floral syndrome makes no sense.

Line 311-312: brightly white tepals ….. Do you mean white ligules?

Lines 437-441: Please improve the wording.

Lines 465-466: You mean that each capitulum was observed every day?

Line 469-470: Again, be careful with the term ‘’tubular flower diameter’’. The term needs to be corrected.

Line 512: What do you mean by ‘’10 × 10 times’’? Also on line 526.

Line 572: Instead of ‘’bright white petals’’ please write ‘’ bright white ligules’’. What is a ‘’capital flower disc’’? (line 573).

Figure A1 (line 592). Photo A1, A2 show an undeveloped capitulum, not flower buds! The same applies to the meanings that follow in the same caption. You confuse capitula with flowers (florets).

Line 596: instead of tongue flower please write ligulate flower.

Line 599: instead of ‘’flower opening’’ I would prefer ‘’flower maturation’’

Author Response

Response to the Review Comments

Dear,

We great thank for your professional comments on our manuscript. According to your suggestions, we carefully reedited and reviewed our paper, the detailed reviewing is listed below.

The authrors present their results on the reproductive biology and breeding systems of two Opisthopappus species in China. I find the manuscript interesting, and their results are promising for publication. However, there are many problems in the manuscript, particularly with respect to the correct use of botanical terminology. In its current form, the manuscript cannot be published. The following comments may help the authors. Apart from my observations, I strongly recommend that the authors study their manuscript very carefully, offer more clarity to their sentences, make good use of the English language and be very careful with respect to the botanical terminology. The latter should apply correctly throughout the manuscript.

Response: Thank your comments. We carefully reviewed the whole manuscript. The botanical terminology of some sentences or words were all carefully reedited.

line14: O. taihangensis has relatively larger, more ligulate, and tubular flowers. I have difficulties to understand this sentence. Does it mean that O. taihangensis has larger ligules and a larger number of florets per capitulum?

Response: Yes. O. taihangensis has a relatively larger capitulum and a greater number of ligules and florets than O. longilobus.

Line 15: characteristics – You are referring to only one character, protandry. Please use singular.

Response: The relative sentences were reviewed.

Line 19-20: albeit certain visiting insects were also observed for the two species. – You mean insects different that Hymenoptera and Diptera or that there are some common insect species visiting the two species?

Response: According the observation, some common insect species can visit O. taihangensis and O. longilobus. And the relative sentences were rewrote. Thank you.

Line 25-26: Meanwhile, O. taihangensis descendants could better adapt to the severe surroundings than its ancestor O. longilobus … - Again, I feel that this sentence needs a more precise and clear meaning.

Response: The sentences of line 25-26 were carefully rewrote.

Lines 68-69: Please provide some more details about the ancestor-descendant relationships between the two species. On which knowledge / investigation do you base your statement? How is this related to the uplifting of the Taihang Mountains?

Response: Thank you much. The lines 68-69 were reviewed.

Line 87: This study not only helps to improve the existing status of these cliff species – Please consider rewording to make the meaning of the sentence clearer.

Response: The relatively sentences were reedited.

Line 91: The Opisthopappus genus are an autumn flowering species – this does not make sense. I propose: The Opisthopappus genus comprises autumn flowering species. During September …..

Response: According your suggestion, we rewrote the relative sentences.

Line 95: the word ‘’two’’ is written twice.

Response: The repeat word “two” was deleted.

Lines 97-101: the meaning is uncertain. Do you mean that the ligulate florets of O. taihangensis are pink when in bud and remain pink while unfolding, turning white the 5th-6th day? Please consider rephrasing the whole sentence.

Response: The sentences in lines 97-101 were rephrased. Thank you.

Line 102: Instead of writing ‘’floral phenology of a single flower’’ please write ‘’ floral phenology of a single capitulum’’. The flowering head of Opisthopapus is an inflorescence (capitulum) that includes both ligulate (ray) flowers and disk (tubular) flowers. When you describe floral phenology (lines 102-110) please be very precise. Instead of using the words ‘’open, begins to open, opening’’ please use the word ‘’mature, maturation’’ etc. You can also write that the connected corolla lobes of the tubular florets split apart exhibiting reproductive organs.

Response: We reedited those words with botanical terminology according to your comments.

Line 114: Instead of ‘’Flower opening model’’ please write: ‘’Developmental stages of capitulum maturation’’ or something similar. Again, the whole process refers to the whole head (capitulum), not to each separate floret.

Response: The sentences were carefully rewrote.

Line 120: Instead of ‘’ diameters of the capitates’’ please write ‘’the diameter of the capitula (or flowering heads)’’. Please also correct accordingly the legend of Table 1. Do the same throughout the manuscript, whenever by the word ‘’capitate’’ you mean the flowering head.

Response: The relative words were reedited and even in the whole manuscript.

Line 121-122: Please write: The number of ligulate flowers per capitulum …..

Response: The line 121-122 has been rewritten, thank you much.

Lines 119-127: Please add ‘’mm’’ where necessary.

Response: The unit “mm” was added.

Line 128: When you write ‘’flower dimensions’’ you mean capitulum dimensions?

Response: The line 128 was rewrote. Thank you.

Table 1: Tubular flower diameter (mm) cannot be 12.18±0.77! Tubular flowers are small, tiny! Please correct accordingly.

Response: The contents in Table 1 were reviewed.

Line 140: Please be careful with the use of the word ‘’monoecious’’. It does not fit here. Please correct accordingly. Also, the term ‘’ the stamens were also five-lobed’’ does not make sense. Each stamen has an anther. The anthers are connected to form a five-lobed tube around the style.

Response: Thank you. The sentences in this section were rewrote.

Lines 170-173: You repeat (written twice) that peak of insect visits was between 12.00 and 13.00.

Response: Thank you so much for your careful check. We have deleted the repeated sentences.

Lines 178-179: Use italics for plant names.

Response: We checked the name of plant and reedited them using italics.

Line 181: Diptera is written twice.

Response: Thank for your comments. The Diptera was delected.

Fig. 3. Please carefully check the spelling of all scientific names. Use authorship for all names or omit authorship for all names (you have put authorship in some names). The same applies to the text (lines 191-210). Please remember that all scientific names are written in italics.

Response: The words of lines 191-210 were carefully reedited.

Table 2. Please correct: Pollen viability of Opisthopappus taihangensis and O. longilobus ….

Response: The contents in the table 2 were corrected.

Table 2. You counted pollen viability at different hours of the day. Which day did you make the counts? The same applies to Table 3.

Response: Dear, we counted the pollen viability at 1, 2, 3, 4, 5, 6, and 7 days during the flowering period. The detailed information was in 4.4 section.

Line 262: ‘’The average diameters of the tubular inflorescences’’ is not very well understood. You mean the diameter of the tubular florets bearing part of the capitulum? Also, the meaning of lines 266-271 is not well understood. Please consider rephrasing.

Response: The relative sentences were rewritten, thank you.

This means the average diameter of the head, which has been modified in the article, as well as in lines 266-271.

Response: This section was reedited based on your comments.

Line 265: Instead of putting a date please put a reference number within brackets. The same on lines 281, 396, 508, 533.

Response: We have added the reference number within brackets. Thank you.

Lines 277-279. Please avoid using the term ‘’population’’ for pollen grains.

Response: Thank you. The word “population” was deleted.

Lines 263-269: Please consider re-wording to make meanings clearer.

Response: The parts of lines 263-269 have been rewritten.

Line 279: Only one ovule was observed …. Please add ‘’per flower’’.

Response: Yes, there is only one ovule was observed in our study, “per flower” was added in the manuscript.

Line 296-299 (and also paragraph 4.6). Please provide some more experiments about germination experiments. What time of the year did you make the experiments? Under which conditions?

Response: Thank your suggestion. This section was rewritten.

Line 304: capitulum size, instead of flower size.

Response: “capitulum size” was instead by “flower size”.

Line 306-307: Moreover, the relatively larger and more complex floral syndromes …. The wording larger floral syndrome makes no sense.

Response: Thank you for your advice. We have rewritten this section.

Line 311-312: brightly white tepals ….. Do you mean white ligules?

Response: The sentences were reedited.

Lines 437-441: Please improve the wording.

Response: The lines 437-441 were rewritten, thank you much.

Lines 465-466: You mean that each capitulum was observed every day?

Response: Yes, we observed every day during flowering.

Line 469-470: Again, be careful with the term ‘’tubular flower diameter’’. The term needs to be corrected.

Response: The term was corrected, thank you much.

Line 512: What do you mean by ‘’10 × 10 times’’? Also on line 526.

Response: Thank for your comments. The relative parts were reedited.

Line 572: Instead of ‘’bright white petals’’ please write ‘’ bright white ligules’’. What is a ‘’capital flower disc’’? (line 573).

Response: We have rewritten this part according to the suggestions.

Figure A1 (line 592). Photo A1, A2 show an undeveloped capitulum, not flower buds! The same applies to the meanings that follow in the same caption. You confuse capitula with flowers (florets).

Response: Based on your comments, the figure A1 was reviewed.

Line 596: instead of tongue flower please write ligulate flower.

Response: The words were instead, thank you again.

Line 599: instead of ‘’flower opening’’ I would prefer ‘’flower maturation’’

Response: “flower opening” was instead by “flower maturation”.

Sincerely,

Yafei Lan and Yiling Wang

Reviewer 2 Report

Wang et al. present the first detailed data concerning the reproductive biology and breeding systems of two perennial, endemic species - Opisthopappus longilobus and O. taihangensis. The manuscript show interesting and important information from evolutionary point of view. The manuscript is well arranged, but needs some improvements.  Below are my comments and suggestions.

Title. Since the species studied are endemic, I suggest to include this information in the title.

I suggest also to add more Keywords, for example: protandry, autogamy… and simultaneously to exclude Opisthopappus and Taihang Mountains, because these terms are in the title.

Introduction

This chapter well introduces to the study problem. The last question given as the study aim is problematic, since studies were conducted in gardens.

Row 69. the florescence should be rather replaced by flowering.

Results

The first subchapter of the Results is too descriptive and should be shortened. For example, values of most flower traits in the text are located also in Table 1.

Rows 123-127. The measures of such parameters as length and width should be given in mm or other units.

Row 95. Remove one of the “two”

Rows 102-110. In the text authors distinguished 9 phases, while on Fig 1. I see 10 phases. Please,  correct it.

Fig. 1. The title of this figure is: Flower opening model. Is it really flower model or inflorescence model? 

Row 132. I suggest to replace Organ size by flower traits. I also propose to include in this Table statistically significant differences between parameters studied. Species names should be given in italic – this should be also corrected in different parts of the manuscript.

Rows 166

Rows 170-176. Could you show visitation activity in numbers?

Row 179. The point should be probably replaced by comma.

Fig. 3. Species names in italic.

Rows 191-210. Again, is it possible to present data in numbers?

Subchapter 2.2. It would be interesting to know visitation frequency of particular insects. Moreover, proportion of insects with pollen (true pollinators) should be given. Additionally, how many individuals of insects visiting flowers were observed?

Rows 212-213. What show these numbers? Please rewrite this sentence.

Table 2. required better explanation.

Tables 1 and 2 could be explained in the same way.

Row 263. Tubular inflorescences or tubular part of inflorescences?

Row 277. Pollen grain population is not adequate term, in my opinion. Population is well defined in ecology.

Row 278. wwas?

Rows 294-295. Did you noted statistically differences between two species in germination rate?

Discussion

In my opinion the results should be better integrated in the context of main problem. Moreover, differences between two species should be accented.  I don’t see sufficient answer for the last question from the Introduction.

Rows 305-308. This statement is not clear for me and requires wider explanation, especially improved adaptation of O. thaihangensis. What indicate larger floral syndromes?

Row 312. Proterandry?

Row 365. There are rather weather not environmental conditions.

Row 370. It seems that the second part of the sentence (which results…) should be removed.

Row 419. Seed mass – I don’t see this data.

What about of possibility of interspecies gene flow, since two species studied are characterized by the similar phenology, some flower traits are also similar and species may share pollinators?

Material and methods

Row 465. In which way the flowers were labelled?

Row 556. “Fifty seeds of each treatment with three replicates were placed in one culture dish” – In which way seeds were chosen for this experiment? Randomly or were selected?

Rows 552-553. 15 individuals were included – could you give info about number of flowers ?

Although, I am not qualified to assess the quality of English, but it seems that it should be  corrected.

Author Response

Response to the Review Comments

Dear,

We great thank for your professional comments on our manuscript. According to your suggestions, we carefully reedited and reviewed our paper, the detailed reviewing is listed below.

Title. Since the species studied are endemic, I suggest to include this information in the title.

Response: The title of our manuscript was reviewed, thank you.

I suggest also to add more Keywords, for example: protandry, autogamy… and simultaneously to exclude Opisthopappus and Taihang Mountains, because these terms are in the title.

Response: The key words, such as mating systems and visiting insects, were added.

Introduction

This chapter well introduces to the study problem. The last question given as the study aim is problematic, since studies were conducted in gardens.

Response: Thank your suggestion, the chapter of Introduction was reedited.

Row 69. the florescence should be rather replaced by flowering.

Response: “florescence” was instead by “flowering”.

Results

The first subchapter of the Results is too descriptive and should be shortened. For example, values of most flower traits in the text are located also in Table 1.

Response: According the comments, the first section of the Results was reviewed.

Rows 123-127. The measures of such parameters as length and width should be given in mm or other units.

Response: The rows 123-127 were reviewed.

Row 95. Remove one of the “two”

Response: The repeat “two” was deleted, thank you.

Rows 102-110. In the text authors distinguished 9 phases, while on Fig 1. I see 10 phases. Please,  correct it.

Response: Thank you so much. The rows 102-110 were corrected.

Fig. 1. The title of this figure is: Flower opening model. Is it really flower model or inflorescence model?

Response: The title of figure 1 was reviewed.

Row 132. I suggest to replace Organ size by flower traits. I also propose to include in this Table statistically significant differences between parameters studied. Species names should be given in italic – this should be also corrected in different parts of the manuscript.

Response: Thank for your comments. We reedited the row 132 and corrected the table 1.

Rows 166

Response: The row 166 was corrected.

Rows 170-176. Could you show visitation activity in numbers?

Response: We added the relative results in the attached table. Thank you much.

Row 179. The point should be probably replaced by comma.

Response: The word was replaced by comma.

Fig. 3. Species names in italic.

Response: We reviewed the species names in italic.

Rows 191-210. Again, is it possible to present data in numbers?

Response: Thank you. The rows 191-210 have been rewritten, and the data was presented in the attached table.

Subchapter 2.2. It would be interesting to know visitation frequency of particular insects. Moreover, proportion of insects with pollen (true pollinators) should be given. Additionally, how many individuals of insects visiting flowers were observed?

Response: According your suggestions, we reviewed the relative section and added the data in the attached table. The detailed information was seen in the second paragraph of 4.3 section.

Rows 212-213. What show these numbers? Please rewrite this sentence.

Response: The rows 212-213 were rewritten, thank you again.

Table 2. required better explanation.

Response: We added the note under the table 2.

Tables 1 and 2 could be explained in the same way.

Response: We added the note under these tables.

Row 263. Tubular inflorescences or tubular part of inflorescences?

Response: Thank your comments, the sentences were reviewed.

Row 277. Pollen grain population is not adequate term, in my opinion. Population is well defined in ecology.

Response: The “population” was deleted in row 277.

Row 278. wwas?

Response: The sentences in row 278 were reedited, thank you.

Rows 294-295. Did you noted statistically differences between two species in germination rate?

Response: Thank for your comments. We reviewed the relative parts in rows 297-298

Discussion

In my opinion the results should be better integrated in the context of main problem. Moreover, differences between two species should be accented.  I don’t see sufficient answer for the last question from the Introduction.

Response: Based on the comments, we rewritten the first section of Discussion, and added the relative explanations about the differences between two species in rows 305-309 and 332-343.

Rows 305-308. This statement is not clear for me and requires wider explanation, especially improved adaptation of O. thaihangensis. What indicate larger floral syndromes?

Response: We have rewritten this part of rows 305-308, thank you much.

Row 312. Proterandry?

Response: Thank you for the above suggestions. We revised the word of “Proterandry”.

Row 365. There are rather weather not environmental conditions.

Response: The “environmental” was replaced by “weather”.

Row 370. It seems that the second part of the sentence (which results…) should be removed.

Response: The row 370 was reedited.

Row 419. Seed mass – I don’t see this data.

Response: The sentences of row 419 were reviewed, thank you.

What about of possibility of interspecies gene flow, since two species studied are characterized by the similar phenology, some flower traits are also similar and species may share pollinators?

Response: Thank your points. The gene flow actually occurred between two species through our previous study. However, this gene flow is limited. The limited gene flow might mainly result from two reasons: 1) The geographical isolation. O. thaihangensis distributes on the southern of Taihang Mountains, while O. longilobus mainly in the northern of Taihang Mountains. Under the natural state, the distribution areas of the two species are not found overlapped. 2) These two species grow in the cliff of Taihang Mountains. Even shared the common pollinators, it is difficult for these insects crossing the steep cliffs. Based on the previous results, we built a common garden to further study if the interspecies cross occurred between O. thaihangensis and O. longilobus.

Material and methods

Row 465. In which way the flowers were labelled?

Response: The row 465 was rewritten, thank you.

Row 556. “Fifty seeds of each treatment with three replicates were placed in one culture dish” – In which way seeds were chosen for this experiment? Randomly or were selected?

Response: This section was rewritten according to your comments.

Rows 552-553. 15 individuals were included – could you give info about number of flowers?

Response: The relative information was added in rows 552-553.

Although, I am not qualified to assess the quality of English, but it seems that it should be corrected.

Response: Thank you much. We carefully rewritten and reedited the whole manuscript. The languages were reviewed by a native English-speaker.

Sincerely,

Yafei Lan and Yiling Wang

Round 2

Reviewer 2 Report

After revision manuscript may be published